# Towards a Digital Ecosystem for a Smart City District: Procedure, Results, and Lessons Learned

**Frank Elberzhager \*, Patrick Mennig, Svenja Polst, Simon Scherr and Phil Stüpfert**

Fraunhofer Institute for Experimental Software Engineering IESE, Fraunhofer Platz 1, 67663 Kaiserslautern, Germany; patrick.mennig@iese.fraunhofer.de (P.M.); svenja.polst@iese.fraunhofer.de (S.P.); simon.scherr@iese.fraunhofer.de (S.S.); phil.stuepfert@iese.fraunhofer.de (P.S.)

**\*** Correspondence: frank.elberzhager@iese.fraunhofer.de

**Abstract:** The digital transformation supports many cities on the way to becoming smarter cities, enabling them to enhance digital processes, care about climate-friendly goals, or improve the quality of life of their citizens. However, such changes usually take place step by step and not in a big-bang approach. In order for the direction of the digital transformation to be defined, it is necessary to know and understand the needs and requirements of all relevant stakeholders who will be affected or are intended to use the new digital solutions. As our environment, a smart city district, is currently under construction, we do not know most of the future stakeholders yet. Therefore, we had to find new ways of eliciting the needs and requirements for digital solutions without knowing, e.g., the citizens who will live in the future district. We show a framework of the procedures we followed, classified into (a) vision and concepts, (b) smart city district digital ecosystem, and (c) dissemination and events. We substantiate the processes with example results and provide a discussion on how we evaluate our solutions with respect to future applicability. Because evaluations are only very limited in our setting right now, we focus on four lead questions to argue why the procedures and results are adequate and share the lessons we learned on this path towards a digital smart city district.

**Keywords:** smart city; city district; digital transformation; digital ecosystem; smart city services; lessons learned; climate-friendliness

## 1. Introduction

Providing high quality of life is an aim that many cities strive for. Quality of life is influenced, among other things, by air quality, traffic infrastructure, and recreational quality in public spaces. Digitalization offers many opportunities to improve these factors. Moreover, the digital transformation also helps to address challenges affecting society as a whole, such as climate change. How can cities become smarter and more digital? Which new concepts and solutions could enhance the efficiency of key infrastructure, utilities, and services to create a sustainable city environment that provides a high quality of life for its citizens? Transforming a city into a smart city is often a step-by-step procedure rather than a big-bang change, both for technical reasons [1] and because skeptical citizens need to be convinced of the benefit of new digital solutions.

We are currently part of a research project where a smart city district is being built. The district should be as climate-friendly as possible. Topics such as mobility, energy, waste management, or smart home are relevant in our project setting. In this project, we are developing digital solutions addressing these topics. Due to the complexity and diversity of the topics being addressed, a digital ecosystem is being created for the district. This ecosystem with its platform establishes the foundation for an extensible and holistic solution offering various domain services and solutions.

Beyond this foundation, we developed a game to inform citizens about new mobility concepts and show them that decisions regarding the implementation of new mobility solutions do not make sense for all types of citizens. Another example of our work in

this context is an app that provides users with hints about how to reduce their energy consumption. Details about these solutions and many others are given in Section 3.

However, as we point out in this article, one of our main challenges is that the city district is currently under construction (see more details on the project setting in Section 2.1). This strongly affects our process for developing, applying, and evaluating digital solutions. One consequence is that there is only a very limited number of citizens or other future stakeholders from whom we can elicit needs and requirements. For example, having other citizens outside our city district serve as substitutes is an option that is reasonable only to a limited extent, since, for instance, requirements for mobility solutions are strongly influenced by one's immediate surroundings. Outside the district, someone might say that they absolutely need a car for shopping because the distance would be much too far to carry the groceries. In our city district, rental cargo bikes are planned and the distance to the nearest supermarkets is extremely short, for some residents only about 100 meters. This means that such concrete requirements are dependent on the people living in the district.

Another consequence is that we have to develop solutions that will mainly be used a few years in the future when most construction work will be completed. However, we cannot foresee the future. We do not know which needs the citizens will have in the future. There might be other solutions that fulfill their needs by the time the district is built. Moreover, technical requirements and restrictions might change.

Having no access to the future citizens and further stakeholders also means we do not have the possibility to test our solution with the target group. Nevertheless, the citizens and their needs should be at the center of all our considerations and digital solutions. Otherwise, digital solutions will not be accepted and used.

In this article, our main contributions are that we show concrete steps and procedures for developing digital solutions for a smart city district and share what digitalization of such a climate-friendly district means under the challenge that the city district is still under construction. Therefore, we have to find ways to draw concrete conclusions even if future citizens and further stakeholders are mainly not known today. We share our experiences on that path and provide insights, ideas, and guidelines that can be applied in similar settings. To make it easier to understand our overall framework, we provide several examples that show the outcomes of our procedures with respect to the three topic areas vision and concepts, digital ecosystem, and dissemination and events. Besides individual solutions, we also show where and how solutions benefit from each other and where synergies can be used. Of course, evaluation of solutions without real citizens of the future city district is very limited, but we sketch some substitutional ways and provide an outlook on how we plan to do future evaluations.

In this article, when we present our steps and give concrete examples of solutions we already developed, we share our experiences reflecting four lead questions:

LQ 1. How to elicit needs and user requirements for digital solutions without knowing the users of the future city district?

LQ 2. How to communicate the vision, concepts, and future solutions without having actually implemented them—in other words, how to build a bridge from today to tomorrow and prepare users for the solutions?

LQ 3. How to deal with the challenge that needs will continue to change until the concrete implementation of digital solutions and usage by citizens in the future?

LQ 4. How to test and evaluate the vision, concepts, ideas, and prototypical implementations without users of the future city district?

Basically, we classified our solutions into three concrete topic areas. We see these as key in such a development towards a digital city district ecosystem and focus on them in this article:

(1)    Vision and concepts;
(2)    Smart city district digital ecosystem;
(3)    Dissemination and events.

The structure of this article is as follows: Section 2 provides a more detailed background on the project setting as well as related work. In Section 3, we elaborate our procedures and present examples according to the three topic areas. Wherever possible, we use the lead question(s) that are suitable in the respective context. Every part is described and analyzed, and a summary and conclusion are given. Section 4 discusses the results and provides examples where the individual solutions profit from each other and where synergies are possible. Section 5 concludes our article with an outlook on future work.

## 2. Background and Related Work

### 2.1. Project Overview

The main goal of the "EnStadt:Pfaff" research project (short: Pfaff project) is to develop a climate-neutral smart city district. Together with eight partners from research, industry, and government, we contribute to the energy transition and address climate protection goals. The project runtime is from October 2017 to September 2022. Our focus is on topics such as energy, mobility, smart home, and community. Specifically, we as one of the partners are responsible for digital topics in this smart city district. These comprise developing and implementing a digital ecosystem that consists basically of a platform and digital services, but also finding solutions for disseminating the topic and providing isolated software solutions.

At the beginning of the project, we defined core principles to which we adhere when developing digital solutions, e.g., that the digital city district platform shall unobtrusively support quality of life and climate protection in the neighborhood, or that people's needs are at the heart of the digital platform. The core of the future smart city district will be a platform with several services [2]. Such a digital platform therefore needs to ensure certain qualities when it is operated, such as security aspects, privacy, performance, or usability. However, besides the current challenge that the city district is still under construction and we have only very limited access to the real environment, we also had to find ways to develop solutions that we can evaluate fast. In an operating platform environment, this is very difficult, as the prototypical solutions often do not have the necessary quality and the main focus is often on getting fast feedback rather than on addressing many other qualities. We had to face these challenges when developing our digital solutions.

This was our starting point: We had the great opportunity to develop concepts and digital solutions for a future city district. The area in question is a former industrial complex of a famous sewing machine company that is being re-developed as a living and working environment. The city district is about 18 ha large and will be a mixed district consisting of, e.g., housing, shops, medical facilities, and office space. Some historical buildings remain, but many buildings and most of the infrastructure will be newly developed. Many citizens know the former company, and quite a few actually used to work there, and therefore the topic is also somewhat emotional in our city. The situation gives us the opportunity to conceive of solutions from scratch instead of having to consider existing circumstances; however, there is also a huge drawback, namely, the great challenge of not knowing who will, e.g., live, work, or shop in the city district in the future and what the concrete requirements of those people will be. Thus, when we think about digital solutions to support future citizens of that district in their climate-friendly behavior, we cannot ask them directly or gather information from them as most are not known at the moment. We therefore needed alternative strategies to be able to develop solutions that we anticipate to be relevant and adequate.

### 2.2. Related Work

The term smart city has been around since the 1990s [3]. In the meantime, smart city projects are no longer purely scientific projects but more often than not projects resulting in persistent outcomes for people's daily life. Whereas many past projects focused on making the infrastructure smart with sensors, newer approaches tend to focus more on creating added value for citizens, companies, and the local public sector through digital solutions.

The work of Giffinger et al. [4] lists dimensions of a smart city: smart economy, smart people, smart governance, smart mobility, smart environment, and smart living. Furthermore, they define a smart city as "a city well performing in a forward-looking way in these six characteristics, built on the 'smart' combination of endowments and activities of self-decisive, independent and aware citizens." This definition was cited and discussed by multiple researchers in the following years [5–8]. Similarly, Yang et al. [9] conclude that there are six application domains for smart cities: Natural Resources and Energy, Transportation and Mobility, Building and Infrastructure, Living, Governance, Industry and Human Resources.

### 2.2.1. The Citizen in the Smart City

Even though various aspects are important for a smart city, a large part of research in the field has primarily focused on technology and not on the people living, working, or shopping there [10]. Therefore, Oliveira and Campolargo [10] introduced the term "human smart city". According to them, a human smart city evolves around a collaboration with citizens, leading to a co-design process where the city administration includes the citizens and businesses in the development of the smart city.

The work of Krijveld [11] sees three levels of citizen involvement in smart city development. The levels lead to more power and higher interaction with the process for the citizens. In the first level, crowdsourcing and crowdmapping are used and the citizens act as sensors of the city. This is a passive creation process. In the second stage, citizens are embedded into a co-creation process. The third level of citizen involvement puts the citizen in the lead of the development. This means that in contrast to the second level, the citizens are not only contributing to activities initiated, e.g., by a company, the government, or a research team, but can also create their own products and services within the smart city. Therefore, Krijveld refers to this stage as self-organization, as the crowd decides and builds their own services.

Tadili and Fasly [12] also investigated citizen participation in a survey. For them, minimal participation is getting informed (which is less than the first level of Krijveld). Participation then increases in five steps to the moment where the citizens are in control of the process. Their study concludes that even though smart city experts are aware that citizen are crucial experts, they are mostly involved in early stages of development and later on for reporting problems. Most smart city projects do not have a dedicated budget for establishing a co-creation process and do not have a long-term vision of how to involve the citizens.

The newer work of Mohseni and Behnagh [13] investigated several smart city projects using Arnstein's ladder of citizen participation [14]. They identified a trend towards giving citizens more power over the development process.

Knowing, on the one hand, that our district will still be under construction over the next few years, and, on the other hand, the importance of citizen involvement led to one of the challenges our project has to address. Therefore, one of our major project contributions was how to involve, inform, and include citizens with regard to the future smart city district and how to get as close to understanding the potential future residents as possible.

### 2.2.2. The Digital Ecosystem of the Smart City

The definition of a smart city according to Mamkaitis et al. [15] is more technology-driven and not citizen-centered. They define a smart city as "an Information and Communication Technologies (ICT) enabled development which extensively uses information as a way to improve quality of life for its citizens and population at large". This means that in the smart city, "sensors, social media, web activities, tracking devices, etc. generate various and large amounts of real-time data" [16] to accomplish this task. This leads to many data points resulting in the situation that Big Data analytics techniques have to be used to process them in the desired way [17]. Nevertheless, many of the ICT solutions built focus only on one or a few aspects of a smart city [16]; e.g., Teslya et al. focus on

mobility [18]. A mature smart city should evolve in several of the areas mentioned by Giffinger et al. [4]. This throws up the need for an integrated perspective on a smart city. Therefore, proposed smart city maturity models like the one of Afonso et al. [19] measure the smartness of a smart city in eleven dimensions ranging from water via health to media.

Developing software systems for the smart city domain is a complex task. Nesi et al. [16] present an integrated multi-purpose platform for this problem. Their focus is on unifying data from different sources and supporting the data value chain. D'Aquin [20] concludes that a potential reference technology stack for a smart city contains "aspects of networking technologies, sensors and physical devices, communication technologies, cyber-physical systems, information systems, data management, federation and distribution, open data, analytics, visualization, machine learning, and many more components". This makes the platform a complex but powerful element of the smart city. The challenge in developing a smart city platform compared to other platforms is that in a smart city [21], the services utilizing the platform are usually from multiple domains and result in different application types.

Recent research shows that a holistic smart city should not only be built on the basis of a platform but should be embedded into a digital ecosystem. According to Koch [22], the central aspect of a digital ecosystem is that multiple organizations collaborate with their technical solutions over a central platform. Due to the complexity of the created solutions, using the platform, eliciting requirements, and of course designing the solutions or services in the ecosystem are challenging tasks. The platform assists here by offering a basic infrastructure for the solutions or core technologies [23]. This could be, for example, a shared authentication and role management system for the ecosystem.

According to Mahesa et al. [24], in the context of smart cities, the platform should allow opening data systems and should enable citizens to give feedback in order to improve the services of the smart city system and the management of the city. According to Abu-Matar and Davies, a central aspect of the platform therefore could be a data integration hub [25].

The challenge of bringing all the different services for our smart city project together and the challenge of building the solutions themselves made us think about the ecosystem service development process and how to ease the prototyping and feedback process for the solutions in our planned ecosystem. Instead of focusing only on the final ecosystem to be built, we investigated the innovation process for creating solutions for a smart city ecosystem. Therefore, a prototyping environment for third parties is a central aspect of our solutions.

### 3. Procedures and Example Results

In this section, we provide our conceptual framework, which follows an applied research approach: We demonstrate our procedures and show example results that were developed during the different steps. We show which procedures we applied and how we adapted them to cope with our project challenges (see Introduction and Section 2.1). We classified our framework into three categories: "vision and concepts", "smart city district digital ecosystem", and "dissemination and events". Each category is motivated in the following sections, followed by examples from our project and a summary that reflects which of our four lead questions were addressed and how.

Figure 1 shows an overview of the different solutions and at which time we worked on them. Due to the different kinds of solution areas, we could not define one approach or one procedure that fits all. Instead, we followed an explorative approach, which is also justified by the innovative character of our research project. However, the individual procedures are partly based on established state-of-the-art solutions, adapted with innovative concepts where necessary.

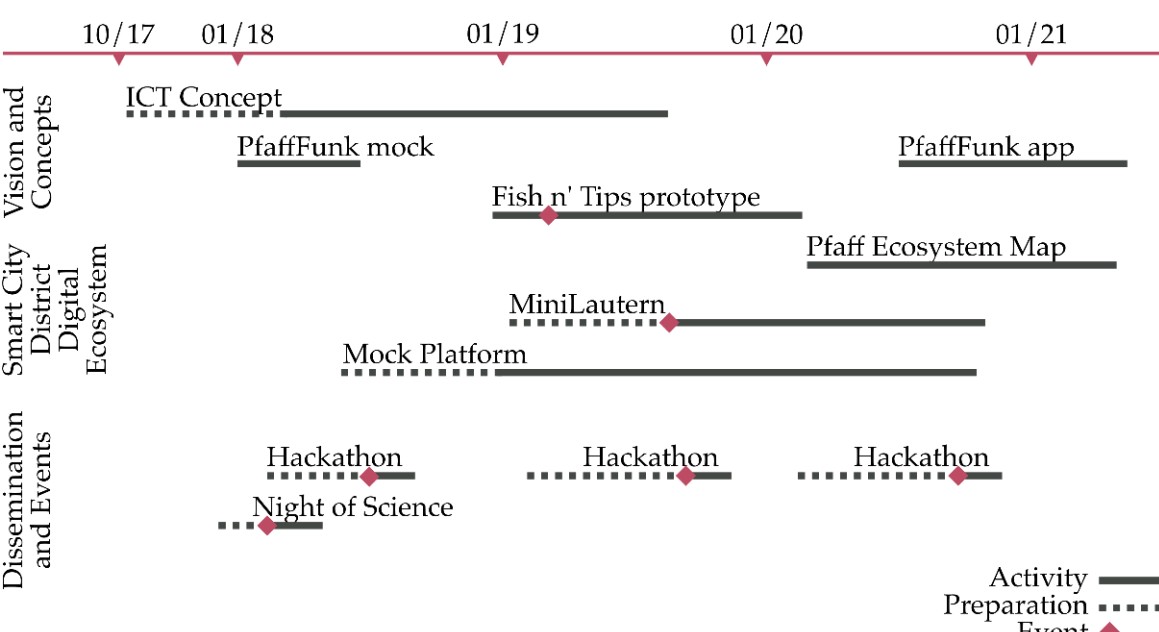

**Figure 1.** Timeline of the different solution directions comprising our framework.

### 3.1. Vision and Concepts

At the beginning of our research project, we had to start with a vision for the future digital smart city district. The goal was to think about the general solution space and make it broad, as well as to narrow it down afterwards to our concrete project setting. This activity required us to find answers to the following questions: How can digitalization support a smart city district? What do we need as infrastructure, who are our stakeholder groups, what services can we offer to them? Two examples are presented next: We worked on an "ICT concept" (Section 3.1.1) during the first two years of our project and used this as a supporting instrument for developing concrete solutions afterwards (we also present some examples in Section 3.1.2). Furthermore, we are developing a so-called digital ecosystem map, which is our means for communicating our digital solutions easily (Section 3.1.3).

### 3.1.1. ICT Concept

Early in the project, we began investigating the potential of digital solutions to support future citizens of the smart city district in living an environmentally friendly life. Climate protection directly refers to the general goals of the project. They are in turn deduced from the German government's 2050 climate protection targets, which are reduction of greenhouse gas emissions by at least 80% to 95% compared to 1990, increase of the share of renewable energies in gross final energy consumption to 60%, and reduction of primary or final energy consumption by 50% compared to 2008.

Within our scope, we want to explore and implement digital solutions that will support future smart city citizens in their daily life in two perspectives. On the one hand, these solutions shall provide value by solving users' problems or addressing their needs. On the other hand, these digital solutions shall motivate, nudge, or support users' behavior that is beneficial for climate protection and reduce, make unpleasant, or impede behavior that impacts climate protection negatively, but without compromising the first perspective.

We identified three major areas of concern for such digital solutions: energy, mobility, and community, and put our focus on user-facing solutions (applications). This does not imply that we completely neglect cyber-physical systems, but rather that the emphasis on users is at the center of our efforts. All proposed applications shall provide value to the future users. If this requires the use of cyber-physical systems, we still call them "applications" to stress the user focus. The first area of concern, energy, refers to all applications that help to reduce energy consumption or increase the use of renewable

energy by smart city citizens. Mobility covers applications that support a reduction of overall mobility or increase the use of climate-friendly mobility means. Community covers applications that support the climate protection targets through changes in the daily life of smart city citizens. Potential solutions can be assigned to one or more of these areas, depending on their focus and design.

As the future smart city district is currently under development, no citizens, i.e., actual users, have been available for eliciting requirements or testing solution prototypes. We developed a set of proto-personas (see Figure 2, cf. [26]) representing the archetypes of future smart city district citizens. Through creativity and innovation workshops, ideation sessions, hackathons, and similar activities undertaken with researchers and local citizens alike, we have developed several different solution ideas for each of the major focus areas (energy, mobility, community) that match the expected needs of the proto-personas as well as the climate protection targets.

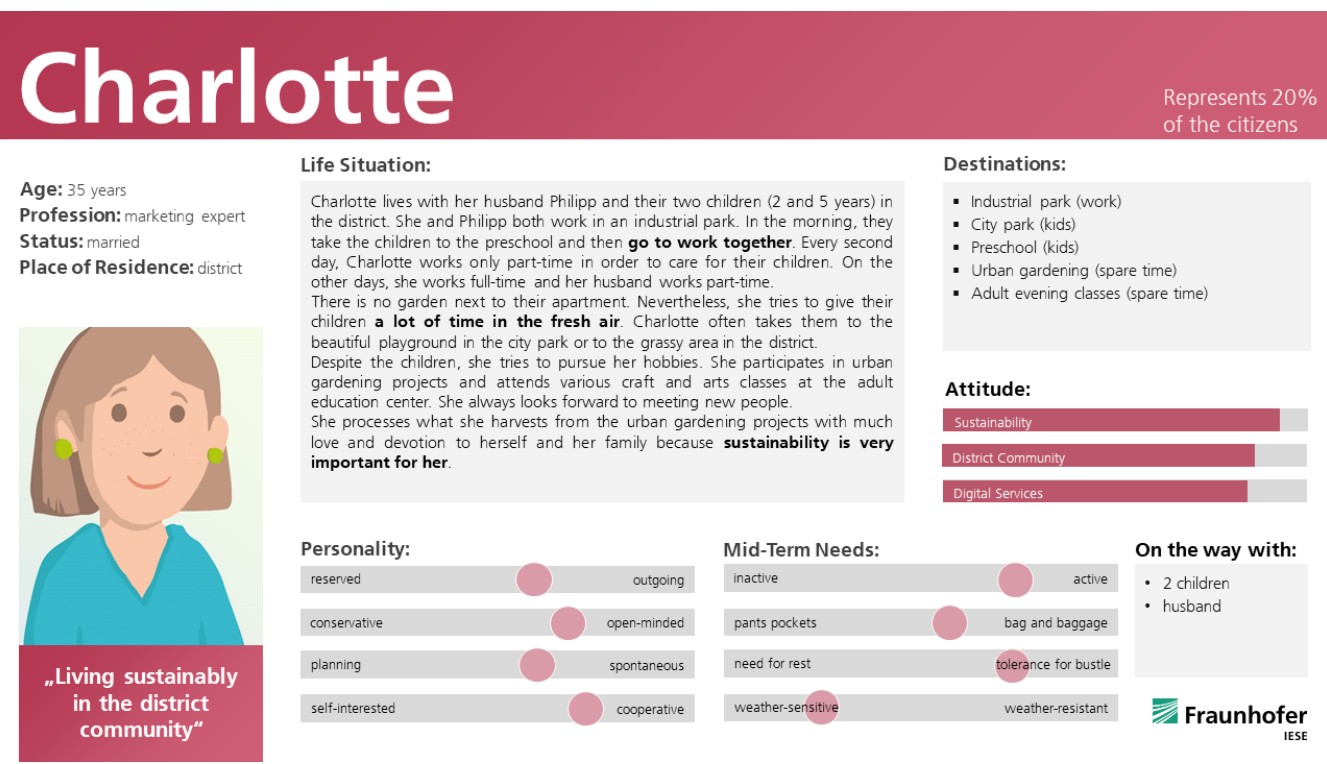

**Figure 2.** Sample image of a proto-persona used in the ICT concept.

Yet, how does one communicate these ideas for potential future applications, evaluate their intended impact, and gather feedback from users who would be affected by these solutions in the future? In the early project phase, we did not build tangible prototypes of suggested applications because we first needed to explore the general direction of potential future solutions before investing in creating high-fidelity prototypes. Our goal, at that project stage, was to communicate our general vision of the digitally supported climate-friendly daily life in the smart city district to the general public.

One way we did this was by releasing the "ICT concept" document (see Figure 3). The document covers descriptions of the project's goals, the vision for a digitally supported smart city district, our approach and way of working in the project, the stakeholders we identified using classical requirements engineering (RE) methods and the proto-personas, the potential solutions and their intended technical resolution, and an outlook towards the digital ecosystem's business model opportunities. All this combined makes up our vision of the digital smart city district. It is more than just the digital applications. Each part of the document's content is described in simple, non-scientific language and enriched

with informative and decorative figures to increase accessibility for a broad audience. We focused on the main ideas of each topic, not losing ourselves in lengthy descriptions and keeping each part concise.

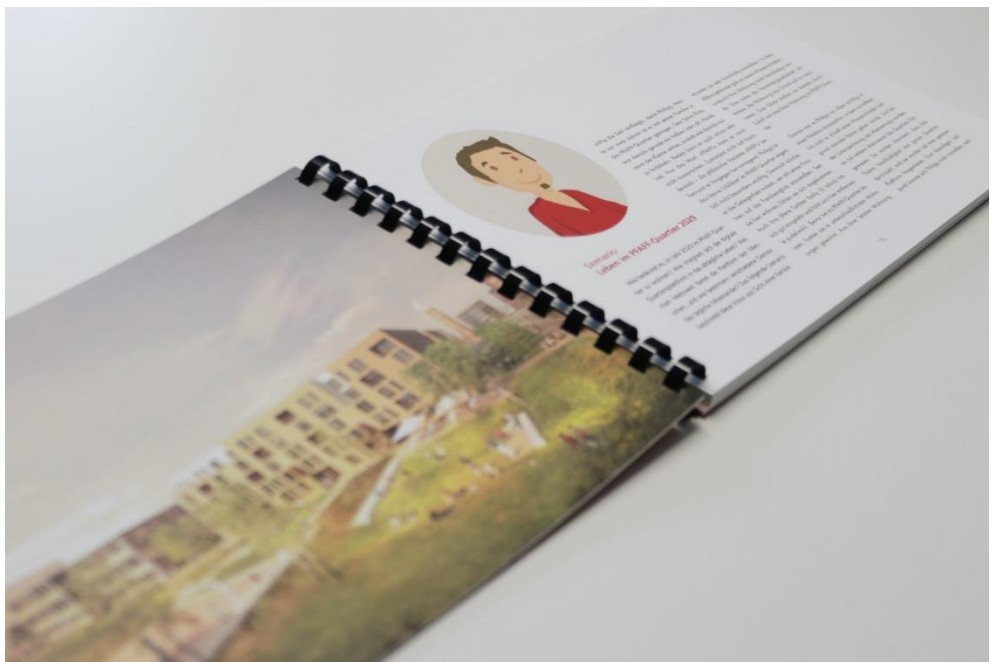

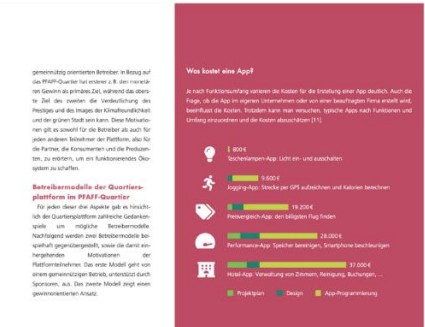
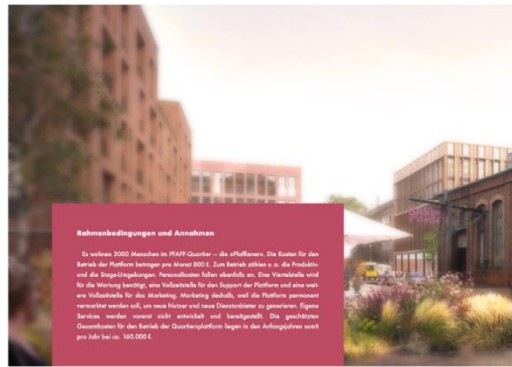

**Figure 3.** (**Top**): Picture of the printed ICT concept showing the illustrated scenario describing life in the future smart city district. (**Bottom**): Two sample pages (in German) from the ICT concept showing the illustrations and visualizations used in the document. To take a look at our representation, we recommend examining the German version of the document available as a pdf file at: https: //oc.iese.de/index.php/s/Rwa3YBHa95iHrVV (accessed on 10 May 2021).

With regard to our lead questions, the ICT concept fulfills a special role compared to other activities presented in this work. It summarizes and abbreviates the key results of requirements engineering activities and other related activities, aiming to make them tangible for a broad audience that, for the most part, is not immediately affected by any of the project's outcomes. LQ1 is only slightly affected by the ICT concept, as it does not represent requirements engineering activities by itself, but summarizes and displays them and their results. However, lead question LQ2 can be discussed here.

The whole document has been designed in a way that increases appeal and accessibility. The document is printed on thick paper (see Figure 3), which is appealing when touched to increase the chance of people picking up the document and reading it. We acknowledge the irony of printing a document on thick paper for communicating the vision of a climate-friendly smart city district. However, in this case, we valued the importance of

communicating the vision effectively (i.e., the long-term effects) through a small print run over the short-term reduction of greenhouse gases achieved by not printing this document. In addition to the physical copies, the document is distributed online. Readers can access the document whenever they like, increasing the chance of it being read. Overall, we conclude that such a document is well-suited for communicating the general vision and idea of digitally supported smart city districts, as it reaches citizens and future users by reducing the obstacles for them to engage with such a topic. The ICT concept is less suited for creating a deep understanding. Hence, with regard to lead questions LQ3 and LQ4, this document does not provide any support on its own but may be used in conjunction with other activities (e.g., citizen events, backchannels for collecting feedback).

Creating the document required a significant investment of time, as not only the content needed to be transformed into simpler language, but also the visualizations and illustrations had to be drawn. Additionally, the layout process itself took time. Nevertheless, we regularly use this document as the basis for discussion and ideation activities. Hence, we conclude the lesson learned as follows: If one has the capacity and required capabilities (skills, software) for creating such an accessible document, it is worthwhile doing so if it is to be used not only for communicating the vision but also as input for later activities with the general public, as it makes the project's goals and the topic of digitalization tangible.

### 3.1.2. Digital Services Prototypes

The digital solutions we investigated in the ICT concept aim to support future smart city district citizens in integrating climate-friendly actions into their daily life and reducing harmful activities. In addition to exploring digital solutions in that vision document, we created prototypes of two different applications.

The first one is called PfaffFunk and serves as a sort of social network for local communities. We expect this application to support community building and nurturing in the future. This application's value proposition is to easily connect citizens of a certain area (e.g., the smart city district) with each other and provide a safe space for gathering and exchanging information, chit-chatting, and making arrangements. Yet, how would this application support the climate protection goals addressed in the smart city district? It lies within the aspect of building a community of like-minded individuals that all work together to reduce the overall energy consumption of the smart city district and their own carbon footprint. This is only an indirect form of support, but we have experienced similar community-building effects in another domain, smart rural areas, where a comparable application is being applied in real life. The application prototype is fully implemented, is available as native iOS and Android applications, and supports all features mentioned above, even though the local relationship is not yet present due to the smart city district not being established yet. Figure 4 shows screenshots of the current state of the application prototype.

The second prototype digital solution is called Fish n' Tips and shows the vision of an interconnected application that interacts with other digital services and status information regarding the future smart city district. Its value proposition is that a personal coach gives appropriate individual hints to smart city citizens at the right time. These hints are intended to help citizens reduce the climate impact of their actions. For example, a user might be notified with a hint when they leave a window open for a longer time period while the heating is turned on. The information about the user's flat is derived from smart home appliances that will be connected to the digital platform in the future smart city district. The prototype only supports a limited number of hints but can be used to communicate the general vision and idea. It is available as a web application and supports the use of dynamic data for generating the hints; hence, it can be used for truly interactive Wizard of Oz-style user tests. Figure 5 shows screenshots of the current state of the application prototype. These prototypes in and of themselves serve as manifestations of concrete aspects of the general vision of the digitally supported climate-friendly smart city district.

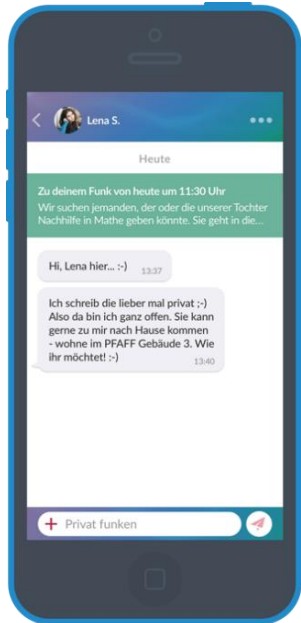 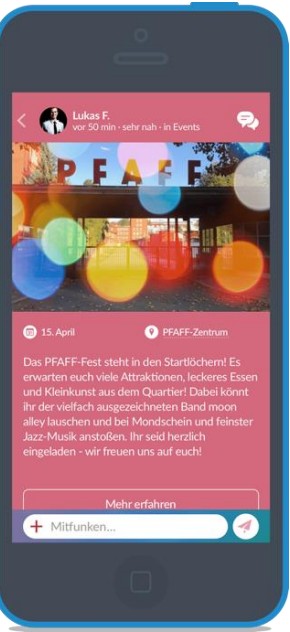 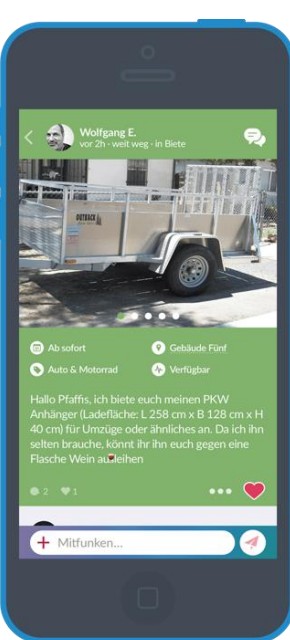

**Figure 4.** Impressions of PfaffFunk: chat (**left**), news (**middle**), and sharing (**right**). The application follows typical patterns of other social networks and chat applications.

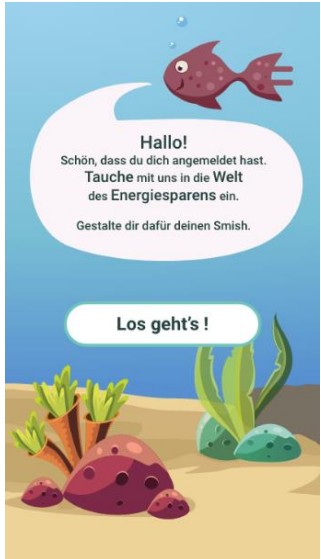 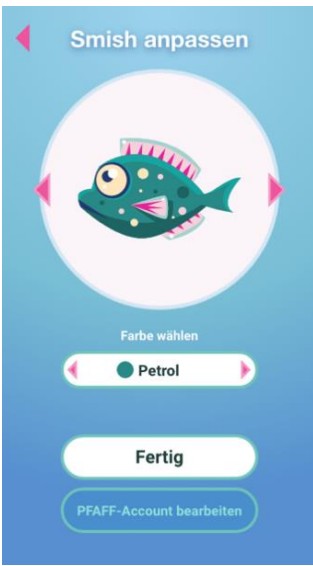 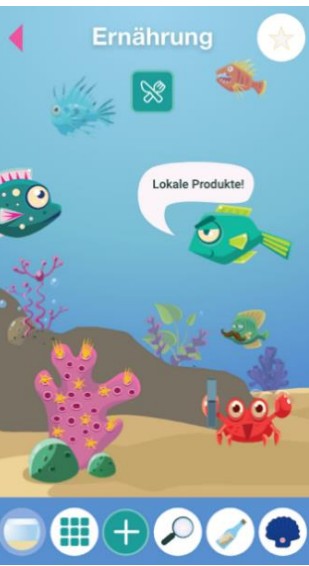

**Figure 5.** Impressions from the Fish n' Tips application. (**Left**): Start screen with a greeting to the user. (**Middle**): The user selects the personal pet (the "Smish") and a color for it. (**Right**): The Smish explores the "sea", looking for hints on how to act in an environmentally friendly way; e.g., the hint shown says "Buy local products".

On their own, the digital solution prototypes only show a small slice of the overall vision for the digitally supported smart city district. Nevertheless, these are valuable tools regarding lead questions LQ2 and LQ4. Tangible prototypes, by their very nature, are helpful for communicating and testing concepts and future solutions without completely implementing them. A prototype is built to serve a certain purpose; e.g., a technical demonstration serves the goal of understanding a certain technology or framework or proving that a certain feature can be implemented. The digital services prototypes are built to serve the goals of communicating the vision of the digitally supported smart city district. We selected the prototypal features to support this goal (e.g., the type of hints given by the

Fish n' Tips application). With regard to lead questions LQ1 and LQ3, we see the digital services prototypes do not provide further support.

### 3.1.3. Pfaff Ecosystem Map

During the project, we had several points where we faced the challenge of communicating and explaining our ideas and what a digital city district means to different stakeholders. From our experience, this gets more challenging the less technological affinity people have, but sometimes this was even challenging when talking to people who do have a general technical understanding but are not deeply involved in digitalization topics. However, this is a very critical and important aspect from our point of view: explaining solutions so that others understand them, see the benefits, and are able to apply them.

In order to have a way to explain the big picture of a digital city district ecosystem on the one hand, but also details about technical solutions on the other hand, we created a so-called smart city district ecosystem map (Pfaff Ecosystem Map). This is a digital visualization of how we understand the digital smart city district. There are several views on this map, but it starts with a high-level view showing streets and buildings of the district and many interaction points for the user. The user can then dive into those parts in which they are interested and see more information when they zoom in. The map itself can show different aspects, for example, concrete services shown as apps, data flow, or technical details about the infrastructure. Such an instrument can be used to actively present the digital solutions to users, but it can also be used directly by the users to explore the map. First ideas how this can look like are shown in Figure 6.

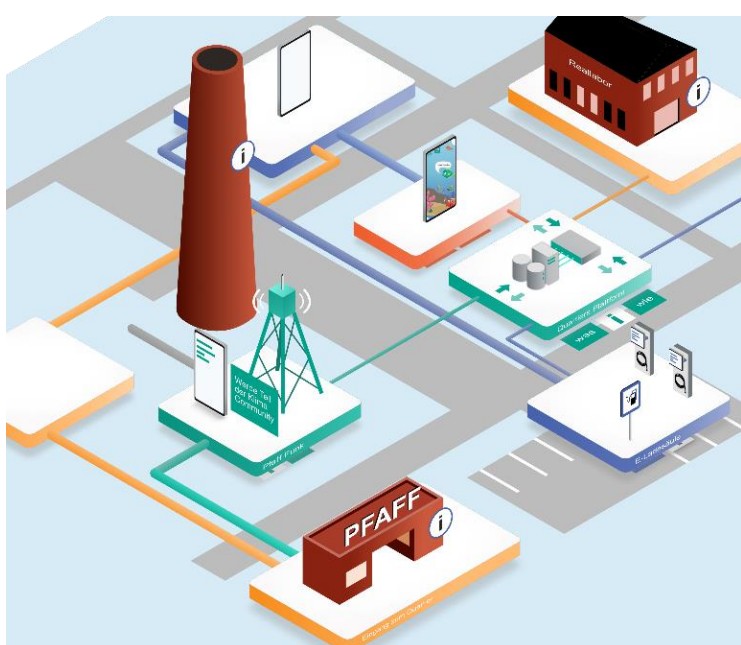

**Figure 6.** Initial prototype of the Pfaff Ecosystem Map visualizing the digital services and their interplay.

In addition, we are also thinking about using virtual or augmented reality (VR/AR) concepts. Of course, as we are currently not in the district itself due to construction work, and therefore we cannot use those concepts there. However, we can already test different tools and technologies to prepare for future VR or AR applications.

### 3.1.4. Summary and Conclusion

In our project, everything started with general concepts and ideas. This took some time. We needed to understand what digitalization of a smart city district is all about, who our stakeholders are, what their needs are, and what potential solutions could be.

We needed to think about the different procedures we want to apply to achieve concrete and correct solutions, as well as about how to share our ideas and knowledge with the citizens. Through various workshops, literature and state-of-the-practice searches, expert interviews, and many discussions (LQ1), we arrived at an ICT concept that summarizes many of our ideas (LQ2). Part of this concept was early mock-ups and visual ideas of services, some of which were developed into prototypes in the subsequent project phase. This brought us to the situation where we had a (theoretical) concept on the one hand and concrete individual solutions in the form of prototypical apps on the other hand (LQ4). However, the bridge in the middle was still missing. We still needed to make it more concrete what a digital ecosystem as a whole means. Therefore, we are currently creating what we call a Digital Ecosystem Map to visualize the idea and the solutions in different ways (LQ2).

### 3.2. Smart City District Digital Ecosystem

The second part relates to concrete services and our technical infrastructure. Two main parts are in the center of what we call a smart city district digital ecosystem: The first part is a smart city district platform. This platform ensures that all services are running, that basic services such as single sign-on are offered, and that the required software quality is ensured. However, we experienced that an operative platform is sometimes not adequate for testing early prototypical solutions. Therefore, we developed a mock platform in addition. This mock platform gives us the opportunity to test new ideas and prototypical services very fast without considering qualities such as security or performance, which would be needed for an operative platform. The second part is a set of concrete services, developed as web services, apps, or webpages, for example. We start with one example service (Section 3.2.1) and provide details on our mock platform afterwards (Section 3.2.2).

### 3.2.1. The MiniLautern Game

Instead of creating a mobility service that will be out of date in a few years, we decided to pave the way for future mobility services. Our goal is to support citizens on their way to accepting more climate-friendly mobility measures and adopting climate-friendly mobility habits. We defined five helpful milestones on that way:

1. Citizens share the attitude that novel mobility measures are desirable.
2. Citizens feel able to use novel mobility measures.
3. Citizens have the intention to change their mobility habits.
4. Citizens try new mobility measures for the first time.
5. Citizens establish new mobility habits.

These milestones are inspired by the Theory of Planned Behavior [27]. For the first milestone, we developed the game MiniLautern, which we will describe in more detail next. To achieve the second and the fourth milestone, we suggest offering offline events such as guided cycling tours with bikes from the local bike sharing system. In this way, new users could ask for help when facing a problem with the system. Moreover, we designed two apps, one called KLaus and the other Fish n' Tips (the latter was introduced in Section 3.1.2). KLaus is a homepage that informs about novel local mobility measures including real persons' experiences. For instance, it explains how to use e-scooters and how the local bike-sharing system works. The content should mainly stem from the users themselves. Fish n' Tips makes suggestions about which mode of transport is best for which trip and which is most environmentally friendly. For the fifth milestone, we developed Luba, an app supporting a competition between teams. The participants receive points for using environmentally friendly transportation modes. The competition should last for several weeks so that the participants can establish new mobility habits.

The basis for the game MiniLautern and the homepage KLaus was created during a Design Sprint [28]. Over the course of five days, an interdisciplinary team interviewed experts in the field of mobility and citizens of the city of Kaiserslautern, generated multiple ideas and concepts, and put them together in two prototypes. Involving the experts and

citizens, we got an idea of the citizens' current attitude and requirements on mobility measures. After the Design Sprint, we continued developing the game; we elaborated the game concept and developed a web app.

MiniLautern (short for tiny Kaiserslautern) is a single-player game with citizens of Kaiserslautern as its target group (see Figure 7). The player should improve the district by establishing novel mobility measures in the smart city district in Kaiserslautern. The player can choose from a list of measures (e.g., reducing the number of parking lots, establishing mobility stations with rental bikes, cars, and scooters or a coworking space). The challenge is to select those measures that have the most positive effect on the quality of life and the individual happiness level of the residents and the least negative impact on the environment.

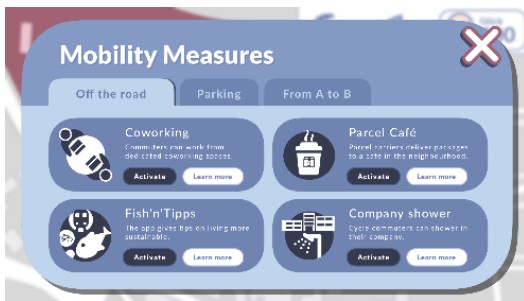 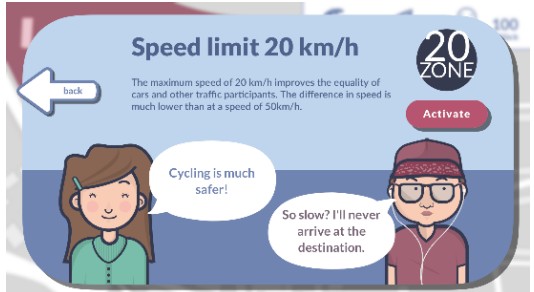

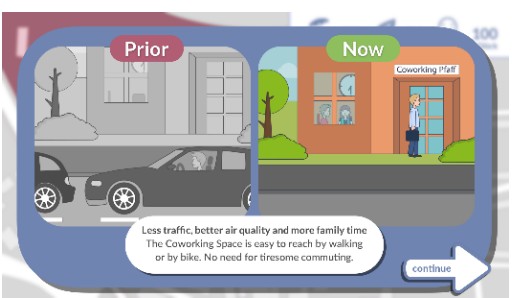 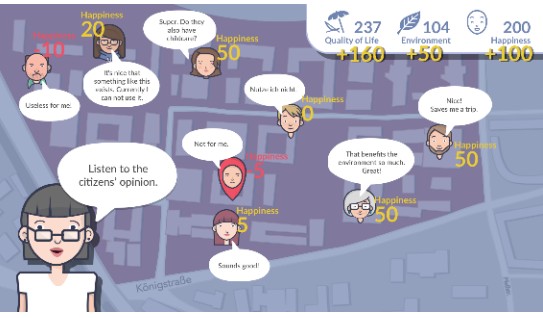

**Figure 7.** Screens of the game MiniLautern. (**Upper left**): Overview of mobility measures. (**Upper right**): Detailed view on a measure with a pro and contra argument. (**Bottom left**): Explanation of the effects of a measure on the environment and on quality of life. (**Bottom right**): Explanation of the effects on individual happiness.

In each round, a player can elicit the citizens' needs by reading about regular destinations, attitudes, and the living situation of a certain number of citizens. Then, the player selects a mobility measure and receives feedback regarding the impact on environment, quality of life, and happiness. Depending on how positive or negative the impact is, the player earns a certain number of points or gets them withdrawn. Most mobility measures in the game lead to a positive impact since this reflects the scientific forecasts and since we aim to establish or strengthen a positive attitude towards novel mobility measures.

We placed the focus on promoting acceptance instead of creating a new mobility service for the following reasons:

- First, the mobility market is currently undergoing major changes, at least in Germany. In cities, there is a shift away from privately owned cars towards public multi-modal mobility services. This upheaval is being accelerated by legal regulations that promote environmental protection, for instance by carsharing, electric cars, and the reduction of parking spaces. Changing legal regulations and new technical possibilities regarding vehicles, such as autonomous vehicles, increase the difficulty to predict the mobility market. Moreover, start-ups as well as well-known automakers such as Volkswagen and BMW are experimenting with novel mobility services themselves. It is difficult

to identify gaps in the market that should be supported by publicly funded research projects such as our project.

- Second, mobility does not end at the borders of the district. Most trips will lead the residents to destinations outside the district. However, solely the district is the subject of our research. This limits the solution space for novel digital services.

To sum up, developing digital mobility services that will fit into a future multimodal mobility system and will fulfill future mobility needs is a huge challenge.

We assume that our game is a suitable way to create a positive attitude towards novel mobility measures and therefore promote their acceptance. By playing the game, citizens learn about the mobility measures and their potential for the smart city district. We assume that a game is an unobtrusive way to get in touch with the measures even if the players have low motivation to learn about the measures. Thus, the game is a suitable way of disseminating the vision of the smart city district (LQ2). Another advantage of the game is that the players have to elaborate on the effects of the mobility measures, which is one prerequisite for building a strong attitude according to the elaboration likelihood [29].

The game includes various game mechanisms. For instance, there is a high score for players who like to compete with others. Some players are motivated by learning more about mobility measures [30], while others enjoy helping the fictional citizens. Addressing several player types increases the likelihood that the game will appeal to many representatives of the target group.

As mentioned before, the players can inform themselves about the mobility needs of fictional citizens in the game. The fictional citizens represent various groups of the population, including families and students. At the beginning, we planned to describe future life situations since the future mobility measures are meant to suit these future life situations. For example, in the future, more and more people will work at least several days per week from home, making commuting to the workplace obsolete but increasing the demand for coworking spaces and carsharing. However, we also want the players to emphasize with at least one fictional citizen so that they feel more engaged. Therefore, the fictional citizens' life situations reflect current life situations and not future ones. A persona template [26] was used as a guideline for creating the description of the fictional citizens. On the basis of the template, we are easily able to create new descriptions (LQ3). Moreover, the game is technically designed in a way that makes it easy to replace text. In this way, we made the game future-proof, so that it can even be applied right after the district has been built, which is relevant as not all mobility measures will be implemented right away. Furthermore, new mobility measures could be included in a similar way (LQ3).

Presenting the mobility measures as part of a game allows us to even present measures that might not be realized. Since it is difficult to foresee the future, we cannot be sure which measures will indeed make it to the district, even if the project members work hard on their implementation.

The game will be part of an exhibition in the district. At the exhibition, we want to get in touch with visitors and receive their feedback about the game and the mobility measures. Some measures cannot be changed anymore at this time, such as the low number of parking spaces, but others such as sharing systems for bikes could be adjusted to their needs.

We advise project teams who consider a similar game to:

- Identify the key message you want to convey so that you stay focused when creating the game concept.
- Learn about typical game and gamification patterns so that the game captivates players from beginning to end.
- Learn about player types, their motives, and their favorite gamification patterns (e.g., as provided by Marczewski [29]) so that the game appeals to many people.
- Learn about psychological theories regarding behavior and attitude change (e.g., Theory of Planned Behavior [27], Elaboration Likelihood Model [30]) so that the players can be persuaded in the most effective way.

- Think about a setting that encourages players to be open-minded towards the content of the game (cf. Ellen Mask).
- Think about a way to take critical voices regarding the content and core message serious so that even these persons engage in the game.
- Think about a way in which the players could personally relate to the game and the mobility measures so that they deal more intensively with the topic.
- Create guidelines for creating and updating content such as mobility measures and mobility needs, and create a flexible architecture of the game so that the game can be adjusted to future circumstances.

### 3.2.2. Mock Platform

A digital ecosystem for a smart city district accompanies the user throughout the day. This results in high requirements for security, safety, privacy/data protection, etc. However, these characteristics stand in the way of the initial development and testing of new ideas. In order for a concept to be tested in its full scope, it is necessary that the prototype that has been created is also used in the later platform context. If a future productive platform were to be used for this purpose, the effort to dock the prototype onto the platform would be very high, since all technical requirements would have to be met. Since this has to be done for each prototype, development is very costly, which can lead to concepts not being tested in the first place. To counter this problem, we developed a prototyping platform called mock platform.

The mock platform is a platform that is intended to support and thus simplify and accelerate the development of prototypes. For this to happen, it aims to reduce the complexity of the productively used platform in such a way that restrictions that are unnecessary for prototype development are reduced as much as possible. For example, complex authentication mechanisms could then be circumvented. However, it is important that it does not impair the basic functionality of the platform, namely, the linking of services. When developing the mock platform, it was also important to us that new platforms can be easily extended with additional services or new functionalities without having to make major adjustments to the existing system. This should make it possible to use the mock platform for a longer period of time. During the development of prototypes in the context of a smart city district, we noticed that functions such as communication between several applications on a smartphone have to be developed again and again. Since this unnecessarily extends the development time, the mock platform should also provide many basic functions that are needed again and again during prototype development, such as a function for creating users.

In order for the mock platform to meet the above requirements, the communication protocol must be designed to be as simple and open as possible so that any new applications can be easily docked. Therefore, we decided to use an event broker based on the MQTT protocol as the central communication node. This MQTT broker follows the publish-subscribe pattern over which everyone can send and receive messages without a sender having to know the receiver (see Figure 8 top). For this to be done, a message is published on the MQTT broker on a unique topic, which serves as an identifier for these types of messages. A recipient must first subscribe to this topic in order to have this message forwarded. Through different subscriptions, a recipient can easily specify what kind of messages they are interested in and can thus pre-filter the messages. Through this asynchronous communication, one is able to add new senders and receivers easily because they only need to connect to the MQTT broker to participate in the ecosystem. However, not every communication is supposed to be sent via the MQTT broker. Messages should only be published if they have an event character, i.e., if they describe a status change, e.g., that a bus has departed or the streetlights have been switched on. A communication between two applications should only happen in individual cases via separate topics. However, in general, direct communication, e.g., via REST-API, should be preferred because these messages are usually uninteresting for other participants of the ecosystem.

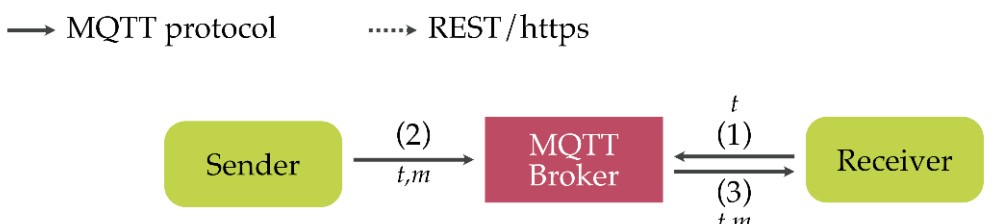

Message *m*: *"currentTemperature: 21.2"*

Topic *t*: *shared-topic/smart-home/building1/appartement1/livingroom/indoor-climate*

(1) Subscribe to topic *t*

(2) Send message *m* to topic *t*

(3) Receive message *m* from topic *t*

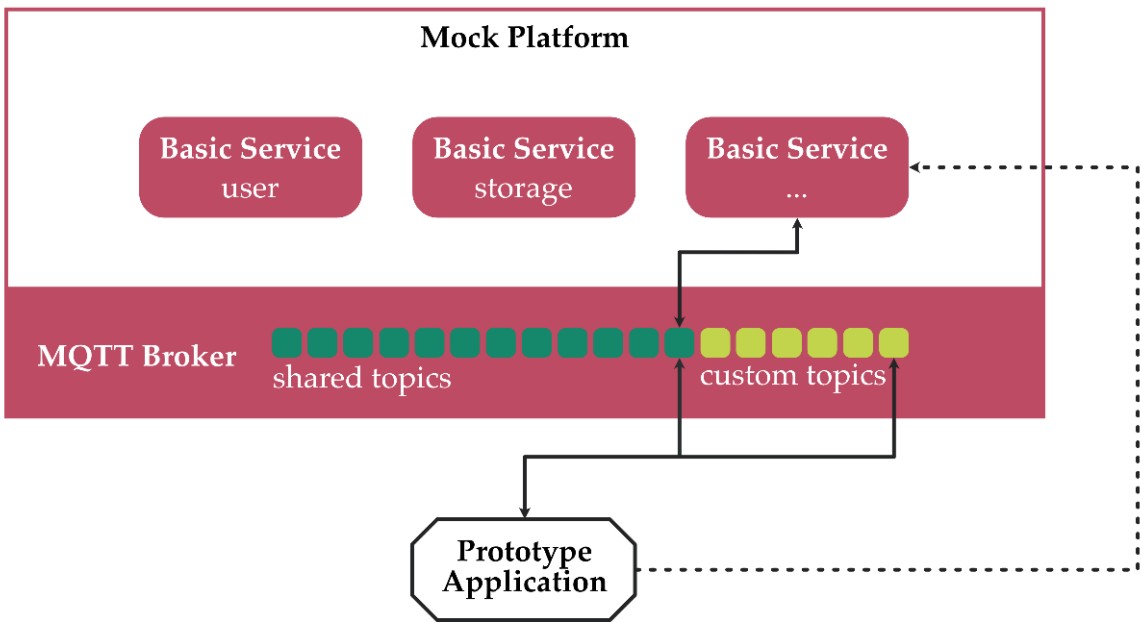

**Figure 8.** (**Top**): MQTT protocol flow. (**Bottom**): General architecture and principal communication between services and applications.

Generally, every sender can decide for itself in which form it publishes its messages and on which topic. However, the recipient must always be informed about this specification so that it can always receive and process the messages. Now it can be the case that similar information, e.g., in the Smart Home area, is published by each Smart Home instance in different ways. In this case, it would be necessary for a receiver that is interested in all Smart Homes messages to be adapted to each sender. To prevent this, we introduced so-called shared topics for several domains. These define for each domain exactly in which form the information has to be published and on which topic (see Figure 8 bottom). Because newly added senders have to adhere to these specifications, already existing recipients do not need to be adapted. Furthermore, new recipients can easily receive and process messages from a variety of existing senders successfully. For this reason, extensibility is easily ensured. The shared topics are predefined by us as the provider of the mock platform and can be extended at any time if required.

Thus, the mock platform enables easy connection of different services through its MQTT broker and the shared topics. As a second component, we extended the mock platform with a variety of services. These provide the basic functions that are needed again

and again during prototype development. For example, there is a user service with which unique users can be created, a storage service that can be used as a central database, and a Smart Home simulator that can simulate a complete fictive Smart Home instance. These additional services are also connected to the MQTT broker and publish various events so that potentially everyone can see what is happening on the platform. If applications want to communicate with these additional services, this is realized via a REST-API, since this represents synchronous communication and the publish–subscribe pattern of the MQTT broker only provides asynchronous communication.

With this ecosystem enabled by the mock platform, many domains such as smart home, power monitoring and distribution, appointment and events, and communication of a smart city district can be covered. In the context of the project, several conceptual ideas or laboratory tests will be developed and executed. For example, in collaboration with the consortium partners, a Smart Home laboratory environment has been created, wherein Smart Home devices are intended to simulate a later inhabited apartment of the city district. The sensors and actuators, such as light switches, thermostats, door and window sensors, or electric shutter controls, collect data. This information is made public to all participants by connecting it to the mock platform. This can add value by automating the processing of another service. For example, energy-saving tips and reminders for the residents could be generated if they forgot to close a window when leaving the building, resulting in unnecessary heat energy loss in winter. On this basis, we created a Smart Home simulator that processes messages on the associated shared topic and displays them in a three-dimensional interactive visualization. Thus, it is already possible for users to visually experience how the future system will behave, without the existence of a real home at this point in time.

The mock platform was deployed at two hackathons, where it showed that it can indeed provide helpful support for realizing or implementing new ideas. Participants rated it as easy to understand, meaning that docking of an application could be done quickly. In addition, the mock platform made it possible to interconnect the individual groups of participants, which was also one of the goals of the hackathon, by making it easy to link their applications with each other. In the future, the mock platform will be extended by further scenarios. The integration of car charging stations or smart street lamps, which can measure, e.g., air quality through a multitude of sensors, are planned.

### 3.2.3. Summary and Conclusions

Our ultimate goal from a digitalization perspective is to have a digital ecosystem that supports different stakeholders in adopting a more climate-friendly behavior. This should be unobtrusive, interesting, and really helpful, to mention but a few of the characteristics of such an ecosystem, at whose center is a platform running various services and apps. As a productive digital environment is not reasonable at this time due to the construction work, we concentrated on a mock platform that enables us to test new ideas and prototypes fast (LQ2, LQ4) and is flexible for future applications (LQ3). Where possible, we connected prototypes to real-world objects, such as Smart Home devices, or plan to do so in the future, e.g., for electric vehicle charging stations or a smart light bulb (LQ4). These concrete examples also make it possible for citizens and other stakeholders to understand how digitalization can support them and where advantages exist for them. Smart Home is an evolving trend, and we are able to show simple application cases to support citizens in their daily life with a Smart Home simulator running on the mock platform (LQ2, LQ4). Another way and example to inform citizens and, ideally, convince them to change their behavior to a more climate-friendly one is our MiniLautern game, where new mobility concepts are explained, but which also shows that change is not easy and does not suit all people (LQ2–Q4).

### 3.3. Dissemination and Events

The third part is dissemination and events. From the beginning, our goal was to communicate with many people. We wanted to spread our project ideas and how digitalization can provide support in many different facets when a smart climate-friendly city districted is created. However, we were also highly interested in feedback from citizens independent of whether they live in the future district or not. We wanted to reduce barriers and fear on the part of citizens and spread a positive understanding of what happens in the actual city district. For this, one early big idea was to organize a hackathon (Section 3.3.1). Having no experience in organizing and conducting such an event, this took a large amount of effort. We later present details on what exactly this meant, but we can also state that this was a great idea in several regards. A second idea that we share here is a kind of workshop—a physical place in the Pfaff district for the discussion of digital topics, the creation of new solutions, and the transfer of knowledge, as well as where people should have an opportunity to come together to advance digitalization topics further (Section 3.3.2). Our third example is from very early in the project where we had the opportunity to participate in a larger event, present our initial ideas, and collect feedback from citizens (Section 3.3.3).

#### 3.3.1. Hackathon

In an effort to come up with a solution approach for creating a large number of potential ideas that reflect the general vision of the future smart city ecosystem, we have planned, organized, and conducted yearly hackathon events. In general, a hackathon (neologism: hacking and marathon) is an event lasting between several hours (synchronous) and several weeks (asynchronous). Participants come together to creatively solve given challenges or problems, with or without restrictions on the technology to be used. Solving the challenges or problems and using the available technology in a clever and creative way refers to the "hacking" in the word hackathon. Hack(ing) is used in the sense of exploratory, playful programming and technology use, not in the sense of computer crime. The "marathon" part of a hackathon is based on the idea of a prolonged event (e.g., lasting 24 h without specific breaks) that challenges participants' endurance and skills. For our hackathons, we called out challenges related to the future smart city district with a focus on supporting or enforcing climate-friendly behavior (climate-friendly perspective). Participants partly provided the user and developer perspectives and we added the technical and organizational perspective through the technology we provided, talks, and facilitator support. Up to now, we have organized three hackathons (2018–2020) with a total of over 100 participants, each with a different focus area, varying target audience, specific circumstances, and high-quality results.

Each of the three hackathon events followed common ideas and guiding principles, although their actual design did vary. Table 1 summarizes the major aspects of the three event instalments, discusses their main differences, and outlines the overall structure and focus of each hackathon. Figure 9 shows impressions of two of the three events.

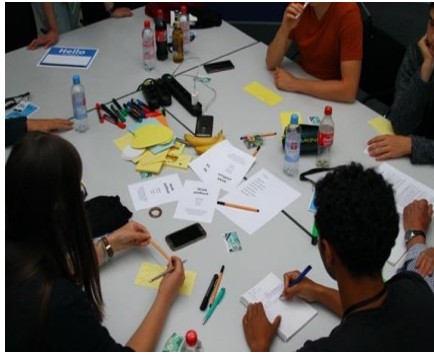 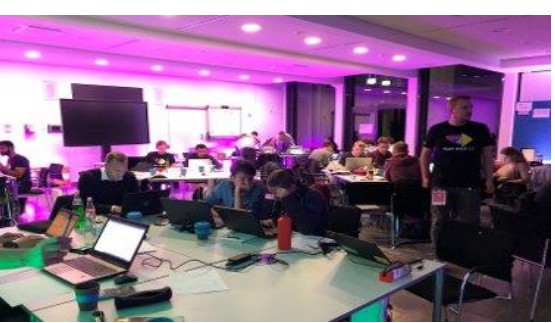

**Figure 9.** Impressions from our hackathons. (**Left**): The first hackathon 2018. (**Right**): The second hackathon 2019.

**Table 1.** Summary of key differences between the first three iterations of the yearly "PFAFF HACK" hackathon.

| | First Hackathon 2018 | Second Hackathon 2019 | Third Hackathon 2020 |
|---|---|---|---|
| **Number of participants** | More than 30 | More than 50 | More than 15 |
| **Intended participants' skills** | No special experience required | Targeted specifically at participants with some development experience or strong interest (e.g., computer science students) | Targeted specifically at participants with some development experience or strong interest (e.g., computer science students) |
| **Teams** | Pre-assigned by organizers to mix skills | By choice of participants | By choice of participants |
| **Challenge/topic** | Concrete challenges regarding energy consumption, mobility, and community aspects of a climate-friendly smart city district | "Smart Green City" as leading topic, no specific challenges | "Smart Green City" as leading topic, participants should aim for interconnected service ideas |
| **Focus of results** | New ideas, new scenarios, paper prototypes | New ideas, running-code prototypes using the mock platform | New ideas, running-code prototypes using the mock platform, interconnected prototypes |
| **Time** | Friday 4 p.m.–Saturday 5 p.m. | Friday 3 p.m.–Saturday 4 p.m. | Friday 3 p.m.—Saturday 2 p.m. (not continuous) |
| **Schedule** | Registration (1 h) Introduction and topics (1 h) Ideation (5.5 h) Intermediate pitch (0.5 h) Prototyping and development (15.5 h) Final presentations and award ceremony (1.5 h) | Registration (1 h) Introduction (0.75 h) Ideation, prototyping, and development (21.25 h) Final presentations and award ceremony (2 h) | Kick-off (1 h) Ideation (2 h) Ideation, prototyping and development, regular talks and interviews in the live stream (20 h) Final presentations and award ceremony (1 h) |
| **Supporting activities** | Ideation session facilitation, technical talks | Technical talks, practical experience report talks | Technical talks (prior to and during the event), interviews, practical experience report talks |
| **Technology and constraints** | None | Mock platform prototype, not mandatory | Mock platform, mandatory |
| **Output** | Three ideas and solution scenarios, low-fidelity prototypes | Eleven prototypes | Three high-fidelity prototypes, partially interconnected |

All three hackathons provided valuable results that served as input for our research project. In total, 17 prototype solutions or solution concepts for challenges of climate-friendly smart city districts of the future were identified and created by the events' participants. The number of solutions scales linearly with the number of participants, but this increases the risk of teams doubling on specific challenges or coming up with similar ideas. Hence, we learned that a decision needs to be made about the size of the event. The organizer needs to provide facilitators and support, guide the teams, and provide food and beverages and the event location (either on-site or the virtual collaboration tools). One cannot deny the effort required for planning, organizing, and conducting such an event. However, we see that the results are worth the effort. The quality of the results varied, depending on the technical skills of the participants. Nevertheless, they all provided a good understanding and visualization of the intended value proposition, making visions of future applications in the climate-friendly smart city ecosystem context tangible.

One may raise the question whether the effort we invested was worth the outcomes. Each hackathon event required serious investment in terms of person-days. The first hackathon event held in 2018 required approximately 40 person-days for event planning,

organization, meetings of the organization committee, event facilitation, and follow-up activities. On top of this, the event required an investment of approximately EUR 2500 for catering and giveaways (e.g., custom T-shirts with the event logo), but these costs were taken over by external sponsors.

We anticipated that the investment in person-days would decrease for the second and third iterations of the event, but due to lessons learned, optimizations, and changing requirements (e.g., due to the remote event in 2020), the effort did not decrease significantly. Would the effort have been better invested in other project activities? In the following, we discuss this with respect to our four lead questions.

Does the hackathon help to elicit user requirements, without actual users, to gather such information without knowing the future citizens of such a city district (LQ1)? Overall, our experience was that the hackathons did not provide major benefits over other activities in this regard, even though the participants might be future users. Our participants were not requirements engineering professionals, nor did the events enforce specific user research activities in advance (e.g., interviews, field studies, observations). However, we did provide the proto-personas detailed in the ICT concept (see Section 3.1.1). We saw that the participants working on the challenges and creating solution scenarios and prototypes just did their best, making educated guesses. However, due to the broad audience and the participants' various backgrounds (i.e., age, social peer groups, country of origin, skills, experience), they reflect a broader part of the general society than the researchers working in this project. Additionally, through talks, documentation, and facilitation activities, we provided input to the participants about the lessons we had learned up to that point. Hence, we conclude that they were able to work with unknown requirements such as other project members. As the participants approached the challenges and problems more naïvely, they did come up with problems we did not consider worthwhile until they showed their solution proposals (e.g., a city gardening support application to increase the consumption of self-grown vegetables and thus reduce the carbon footprint of users, support for home composting through temperature sensors for optimal processes).

In addition, the broad array of solution proposals supports dissemination activities, as different visions of future smart city applications resonate with different target audiences depending on their own requirements and needs. This supports the communication of visions, concepts, and future solutions without having actually implemented them (LQ2). Additionally, the events' participants serve as ambassadors and create a multiplier effect when they talk to their peers about the event. The word-of-mouth effect with respect to the events is hard to measure, but we strive to support this as much as possible. Marketing and dissemination activities are designed to reach an audience as broad as possible through email newsletters, the event website, leading and follow-up blog articles on our institute's website, local newspaper articles, and guest contributions at local radio stations. This supports drawing attention to the project's goals and achievements in general, and to the specific solutions created during the hackathon events.

How well do hackathons support dealing with the challenge that needs will change in the future until the concrete implementation of digital solutions and their usage by citizens (LQ3)? This problem is transformed into a feature during the hackathons, as the goal is not to implement a production-ready solution but a mere prototype to show what could be possible in the future. The underlying concept is to quickly test ideas and create awareness. A large number of participants have been students of computer science and related fields, who might be the future designers of smart city applications themselves, and the awareness for the hackathon's challenges may potentially influence their future activities.

Finally, we want to discuss how to test visions, concepts, ideas, and prototypical implementations without users within the hackathon setting (LQ4). Our events ended with the teams pitching their value propositions and ideas to the other teams and, in the case of the first two events, to the jury. These pitches already served as small tests, obviously with a limited audience, but helped to uncover hidden problems and opportunities. The jury members rated the ideas on the basis of their professional experience and know-how, and

the other teams' reactions supported this by providing a more general perspective. At least one of the teams participating in the second hackathon continued working on their idea and solution, which led to them winning a prize in an innovation competition funded by a German federal ministry.

Hence, we conclude that the hackathon events were indeed worth the effort with respect to our leading questions in particular and the project's goals in general. The fact that the events have changed their shape and have been adapted to reflect the current state of the project is a strength from our point of view, although this increases the difficulty of comparing them. For us, learning from the participants' feedback is as important as being open for innovation and adapting to new requirements and circumstances.

### 3.3.2. City District Workshop for Information and Communication Technologies

In the future city district, we can use a historical building to present our project (and along with this climate-friendly city district topics), but also to create new solutions. In this building, which is currently being restored, there will be an exhibition, but it will also house a so-called city district workshop. In this workshop, we want to elaborate different digital topics with citizens in the area of climate-friendly solutions.

We will especially focus on two main objectives: First, we want to convey and teach (LQ2), and second, we want to design and develop solutions together (LQ1, LQ3, LQ4). Thereby, we want to make the topic of digitization tangible and experienceable. Concretely, as the topic is very broad, we will focus how the digital transformation can support a climate-friendly city district. Actual prototypes and our mock platform may make solutions very concrete. Furthermore, with events such as hackathons taking place in such a workshop room, we can reach people and bring them into contact with such topics.

In essence, we want to address two target groups: technical/digital non-experts and technically/digitally experienced persons. For the non-experts, the threshold must be very low and interest in digital topics needs to be stimulated. Such persons want to learn more and try things out for the first time. Our goal here is to break down barriers and create interest so that they can also take something away and implement it in their everyday life. The experts have two main roles: First, they can teach certain topics to non-experts and help them. Thereby, they support our topics and act as a kind of multipliers. Second, they also want to create concrete solutions. For this, they need interesting opportunities, for example competitive events or equipment that they do not have in their personal environment, but which they want to use to create solutions.

In order to reach both target groups, the city district workshop (a) must be known to them, and (b) has to be as attractive as possible. These two aspects are two concrete "requirements" for us when developing this workshop. A third requirement is (c) to cope with the challenge that it will take several months to really use the building, which is, as mentioned above, still under construction. A fourth demand is (d) to be very flexible, as different kinds of usage of the workshop should be possible. Table 2 shows solution ideas to these major requirements.

**Table 2.** Summary of major requirements and solution ideas of the City District Workshop.

| Major Requirements | Solution Ideas |
| --- | --- |
| (a) Must be known | Presentation over many channels |
| (b) Attractive | Interesting events and application cases |
| (c) Building/room under construction | Mobile character |
| (d) Flexible application | Construction set character |

In order to develop the city district workshop, we had to answer several questions in the beginning: What is our goal? Who is our target group? What material and equipment do we need? What are our major requirements? All of these questions were first discussed internally. As the project has already been ongoing for about three and a half years and

the workshop is being built in our hometown, we already learned quite a large amount about this specific environment and have been able to talk to citizens about the future city district. We furthermore have concrete experience from the past in developing creativity rooms. This knowledge was used by our experts to elaborate the future workshop. We decided to develop our workshop into a "learning and creation room" for digital topics around climate-friendliness, focusing on non-experts (i.e., citizens) and experienced digital people. We decided not to focus only on one concrete room, but to give the workshop a more mobile character so that we can "apply" the workshop everywhere. Of course, not all application cases can be used everywhere, but it is our general intention to pursue certain application cases independent of the location. The advantage of this idea is that we can already use the workshop although the building and the concrete room are not available yet. With this, we can already communicate the topic and talk to citizens and work with them (LQ2).

When the concrete room will be available in the future, the workshop will already be known to a certain extent, as will be the topic of climate-friendliness and how the digital transformation can support this. Of course, to be able to be mobile, our material and equipment has to be flexible. We need cases to transport material, and the equipment itself may not be too large. For example, we plan to have equipment such as a milling machine or a 3D printer. These are examples of equipment that makes the workshop attractive even for experienced persons, as such equipment is typically not available at home, but everything needs to be small and lightweight enough to be able to transport it. Further equipment examples are sensors and actuators for smart homes, moderation material, or even our mock platform for creating digital services (LQ1, LQ4).

We also strive to perform different events in our workshop (LQ1–LQ3). These may be talks, presentations to share knowledge and teach certain topics, but also hackathons that are highly interactive and force participants to create solution prototypes together. Such events will make the workshop attractive and can be used to spread the topic to the public. The participants will gain new knowledge and can then use this to further develop the district. In addition, because of the COVID-19 pandemic, such events can currently not be performed in person, and therefore we are also making video and streaming equipment part of our workshop. This enables us to communicate with citizens and experts during events via video and allows us to create concrete solution prototypes already.

A typical challenge with research projects and outcomes is the question of how to use them after the project is done. In our case, we and other project partners will be able to use the workshop as a dissemination tool, but also for creating further solutions. Due to the mobile and modular character, we can already use the workshop now even though the actual building that will house it is still under construction (LQ1–LQ4).

### 3.3.3. Event "Night of Science"

During the course of the project, we had the opportunity to be part of cultural events offered by the city itself. One such example is the "Night of Science", an event where citizens and researchers all across the city present current topics and discuss them with people visiting the different booths. This was a great opportunity for us. First, we presented early ideas on how we understand the digitalization of the Pfaff city district, gave a brief overview of the project itself, and described the partners involved. Second—and that was the more important part for us—we wanted to use the opportunity to obtain initial feedback from the citizens. As already mentioned, the topic is relevant for many citizens of Kaiserslautern, as Pfaff used to be a big employer and some of the citizens who visited the booth had worked for Pfaff in the past. We wanted to talk to them, obtain ideas from them, and understand their feelings regarding the city district. In addition, we asked for direct feedback and created two posters for this purpose. On these posters, we explained the topics we address in the project so that people could first read about the project. We asked them whether they were interested in talking with us and invited them to participate in a very easy way. We had prepared four tasks for this on a DIN A0 poster:

1. We asked which mobility measures they were currently using most. We prepared answers, and they could post a sticker in the respective parts.
2. We asked what an app looks like that supports saving energy. They could use free text to answer this.
3. We showed a very early prototype of a communication app and they could post emoji stickers indicating how they liked it.
4. We asked very generally what they think about digitalization.

We made sure that the barrier for everyone was very low and did not force anybody to participate. However, many citizens talked to us and gave us early feedback on the posters (Figure 10).

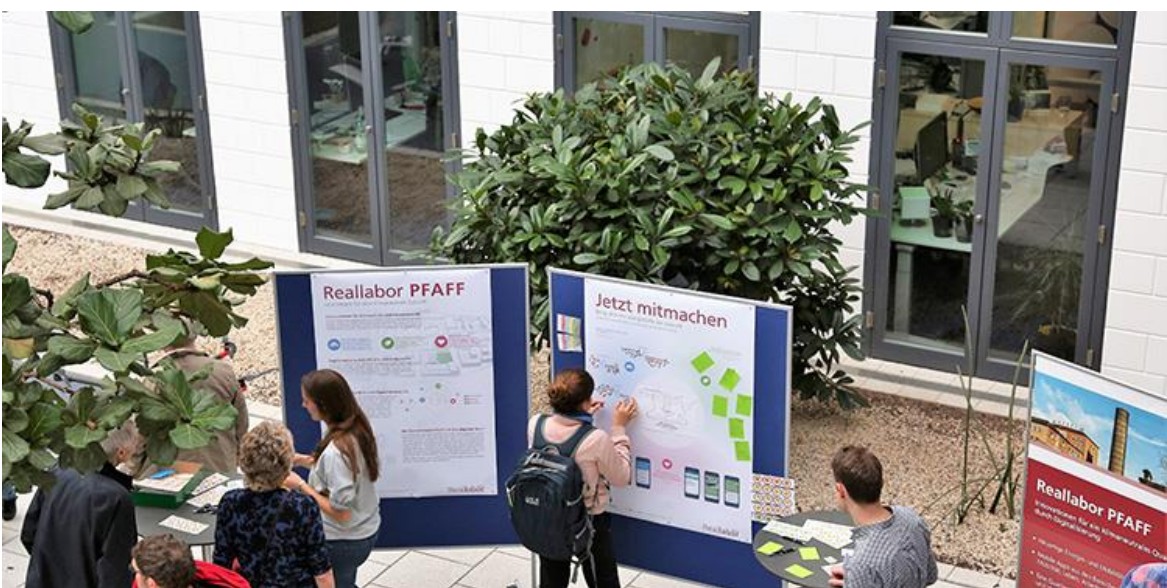

**Figure 10.** Citizens participating at our booth.

Of course, it is largely unlikely that the citizens who gave us input were people who will live in the Pfaff district in the future. Nevertheless, as an early feedback session, this was a great possibility as we did not have to organize a huge event but could simply focus on preparing our booth and being present on the day of the event (LQ1, LQ2, LQ4).

We also want to mention that creating those posters was time-consuming. Questions like "what to present", "how to present", and "how to create easy ways to invite citizens to participate" were not easy to answer, and creating a nice-looking poster also took some effort. However, the results were worth the effort from our perspective.

### 3.3.4. Summary and Conclusions

Promoting concrete solutions, spreading new ideas, and convincing citizens with concrete solutions are some goals of dissemination activities. We do, of course, also use traditional and standard ways, such as a project webpage or articles in the press. However, we did not only share ways of how to inform the public, but also how to gather feedback from them. For this, we organized three hackathons, and plan more in the future (LQ1–Q4). While this was a time-consuming task in terms of preparing and conducting these events, it was very worthwhile with respect to, for instance, including people outside the project, developing new prototypes, and sharing our ideas. A newly developed mobile workshop will be used as a flexible place to again create new solutions, disseminate ideas, and offer events to citizens (LQ1–LQ4). Such events can be used to teach new ideas or topics, but also for discussions, including discussions with citizens. Finally, the third example was the Night of Science event in which we participated and where we also had the chance to talk to citizens and understand their feelings and thoughts regarding the recreation of the

Pfaff district as a climate-friendly district and how digitalization can support this (LQ1, LQ2, LQ4).

## 4. Discussion

### 4.1. Goal and Requirements

The overall goal of our project is to create a climate-neutral city district, and our role as one of the research partners is to develop digital solutions that support this goal (see Section 2.1). Right from the beginning, we started thinking in a technical direction, i.e., for us, the technical core is our digital smart ecosystem platform and services running on that platform, which help citizens and other stakeholder achieve a more climate-friendly behavior. We had already developed a smart rural area platform that we were able to bring into this project, and we managed to adapt some of the existing services from the digital rural context. However, as important as the technical perspective are the people who will live and work in this city district in the future. Therefore, we decided early on to put humans at the center of our work, that is, to derive requirements for digital solutions from humans and to integrate humans into the development and testing of our solutions. Unfortunately, the city district is still under construction, meaning we only have very limited access to citizens who will live and work in the city district in the future, and therefore we needed alternative strategies to come up with concrete requirements for solutions.

Thinking about procedures and results, we found several methods, as described in Section 3. This diversity of procedures is a major characteristic of our framework. On the basis of the three areas (1) vision and concepts, (2) digital ecosystem, and (3) dissemination and events, we followed an applied research approach, i.e., we used established state-of-the-art procedures and adapted them where necessary. For us, our main goals are to support future citizens of the city district and to develop solutions that can be visible, but unobtrusive, that have a positive impact, and that are motivational and simple. Of course, from the viewpoint of the whole project, there are many other requirements to care of, such as legal or process requirements when buying equipment. However, this is beyond the scope of this publication.

### 4.2. Implemented Solutions and Their Limitations

The solutions we create and envision for our smart city district are interconnected with each other. From a technological perspective, the ICT concept serves as the baseline for the technological implementation as well as for public relations purposes. The solution being built brings the ICT concept to life and the concept can be used to increase the knowledge about what a smart city can look like. The solution map is used in a similar way. However, instead of explaining needs and requirements in a general way, the map is used to directly inform the stakeholders about what living will be like when the district is finished. This makes the district more tangible and understandable even before construction is completed and gives people a good understanding of what is to come. In a similar way, the MiniLautern game prepares the mindset of the citizens. Smart mobility concepts are not restricted to smart city districts. Therefore, education about and understanding of these solutions is beneficial for all citizens.

We are also thinking about how knowledge and experience can be shared in innovative ways besides classical channels such as a project webpage or articles in a newspaper. We experienced that these channels account only for a limit number of the people we reach. Therefore, one idea is to make greater use of videos on a webpage or a YouTube channel. Such videos need to be adequate in terms of length, sound and video quality, and content. However, if they are produced in a high-quality fashion, they can be shared with people to inform them about digitalization in the city district, but also train them on certain topics relevant in the city district. One challenge in this regard, however, is to provide new content continuously in order not to lose attraction.

We focused heavily on citizens of the future city district, as we want to support them in their climate-friendly behavior with digital solutions. However, there are many other relevant stakeholders, such as district planners, shop owners, or researchers. During the development of our current solutions, we had them in mind, but did not adapt our solutions very much. However, the workshop, for example, can also be used by planners, and digital services can also be developed and deployed on the mock platform to suit the needs of further stakeholders.

Our hackathons can also be used as a multipurpose tool, as they aim to create and prototype solutions using our mock platform. This gives citizens who are technology enthusiasts an open canvas to create solutions for a district. The created prototypes are then used to communicate with citizens. A similar multipurpose tool is our Fish n' Tips app. All solutions created with the mock platform can themselves be integrated into the system providing tips to citizens on living more sustainably.

While construction work is still ongoing in the district, our district workshop helps to make the digital transformation tangible for everyone. Like the map offering different complexity levels, the workshop offers information for different levels of knowledge. In addition, citizens can also create their own digital transformation projects, possibly using the mock platform or the future real platform for long-term solutions.

This shows that the ecosystem for the district provides an integrated and holistic solution for future citizens living and working there, for the public, and for other interested stakeholders.

All these different methods, procedures, and tools together form our framework. This is an inherent characteristic of our environment: Due to the innovative character of the smart city district and the different requirements of future digital solutions, we could not follow a one-method-fits-all approach. Instead, we identified three different areas of relevance that we focused on (1) vision and concepts, (2) digital ecosystem, and (3) dissemination and events. Within these three areas of our framework, we followed those procedures that fit best and helped us to create concrete solutions. Those procedures are partly state-of-the-art ones such as workshops, interviews, classical software engineering methods, or related work analysis based on literature review procedures, but we also adapt where necessary with innovative concepts such as augmented reality.

*4.3. Evaluation*

As pointed out in our article, the evaluation of our solutions is still limited at this time, and so is the amount of data we have been able to collect. However, some initial evaluation has been possible already, and this will continue and become broader in the future.

For instance, we have evaluated MiniLautern at several points in time. Thus far, we have evaluated the idea of MiniLautern at the Design Sprint, where the game concept was evaluated with high-fidelity mockups and the usability of our prototype was the subject of test mobs. During all evaluation activities, we examined the user experience through observation and interviews. In the future, we want to evaluate whether the vision of the mobility measures is clearly transferred to the users. As mentioned before, the game MiniLautern will be part of an exhibition. At the exhibition, we plan to interview visitors about their experience with the game.

The hackathon events themselves are also subject to evaluation. One can examine the hackathons from different points of view. From the participants' perspective, the hackathons can be evaluated in terms of their fit to participants' expectations and needs. Second, the hackathons can be evaluated from the perspective of the organizers. Here, as well, one can raise the question whether the events' outcomes fit the expectations and needs. Additionally, as we have discussed in Section 3.3.1, the hackathons can be evaluated to check their fit regarding the lead questions and goals of the project—which is tightly connected to the second viewpoint. We have evaluated the hackathons from the first perspective (fit to participants' needs and expectations) through questionnaires that were available for participants to complete during the event (2018 and 2019) or as

an online survey after the event (2020). The physical questionnaires mostly contained handwritten suggestions for improvement. In the online survey, however, the participants answered the question items by marking their answers on a Likert-type scale. In general, the participants' expectations regarding the events were met or exceeded, while certain areas of improvement (e.g., the catering in 2019) were dominant topics in the feedback.

Besides performing dedicated evaluation activities as events, one of our concepts is to continuously evaluate the created services. This means, on the one hand, that the created solutions may include a feedback component for the user of the solution, and, on the other hand, that the mock platform may contain a concept of collecting technical feedback in order to enable the developers to get a better understanding of the service quality.

The feedback component enables users to give feedback at the very time it appears. This can lead to more detailed feedback about an encountered problem as well as about the application itself. The feedback component supports push and pull feedback. Therefore, users can actively start the process of giving feedback whenever they want, and developers can also include certain trigger actions leading to a feedback request to the user. When developing the pull feedback mechanism, we ensured that the dialog does not disturb the user in the task being performed. As Figure 11 shows, the process is easy to use as only emojis are used in the first step. If someone is willing to optionally give detailed feedback, this is done afterwards. This ensures that giving feedback is a task that can be performed quickly.

The feedback component is designed in a way that it can be included with very few steps in mobile applications. Web apps can directly use the API offered by our feedback platform User Echo Service [31]. The collected feedback enables developers to identify new ideas, problems, and delighter features in their products.

### 4.4. Threats to Validity

As we were not able to provide larger evaluations due to the construction situation in the city district, one may debate to what extent discussing threats to validity makes sense. From our point of view, they are reasonable with respect to judging the whole situation and our procedure and example solutions.

In this regard, one threat to validity is whether we started correctly, i.e., whether we elaborated the starting situation well enough. This means, for example, whether we considered all relevant stakeholders, whether we assumed the right kind of mixed city district, or whether we anticipated the future application of digital solutions. Of course, it is impossible to correctly think about everything that is going to or may happen in the future. However, we made sure to be very flexible in what we do and did. For example, we were aware that we need simple solutions, that we need to gain attention, and that we need to convince other people with our solutions. Therefore, one concept was to create a game for this. There may be other games or adaptations of ours in the future, and therefore we ensured with our procedures that this is possible. The current solutions are only examples in this regard that can be used if they fit, but if they do not, it is easy to follow our procedures to create new solutions that fit better. With the experience gained, this has become even easier now.

Another threat is whether we had the right people in mind when creating our solutions. Of course, we want to address various different stakeholders who will be a future part of the city district. We also elaborated potential stakeholders of the district so that we had an idea who might be living and working in the future city district. The exact segmentation of people is still unknown and has also already changed to a certain extent. On purpose, we started focusing on citizens as such, as we know they will be part of the district if they live there, for example. Citizens are very diverse, but one characteristic is that they need to understand what we are doing with our digital solutions, they need to know them, and they need to be convinced by them. Of course, we might not be able to convince all citizens in a first step, but we can take a first step towards achieving these goals. That is what we started: with the game, but also with the hackathons and other events and solutions. We

were also able to test these solutions in small parts with citizens and gain feedback, and the experience we gained from these events was very positive.

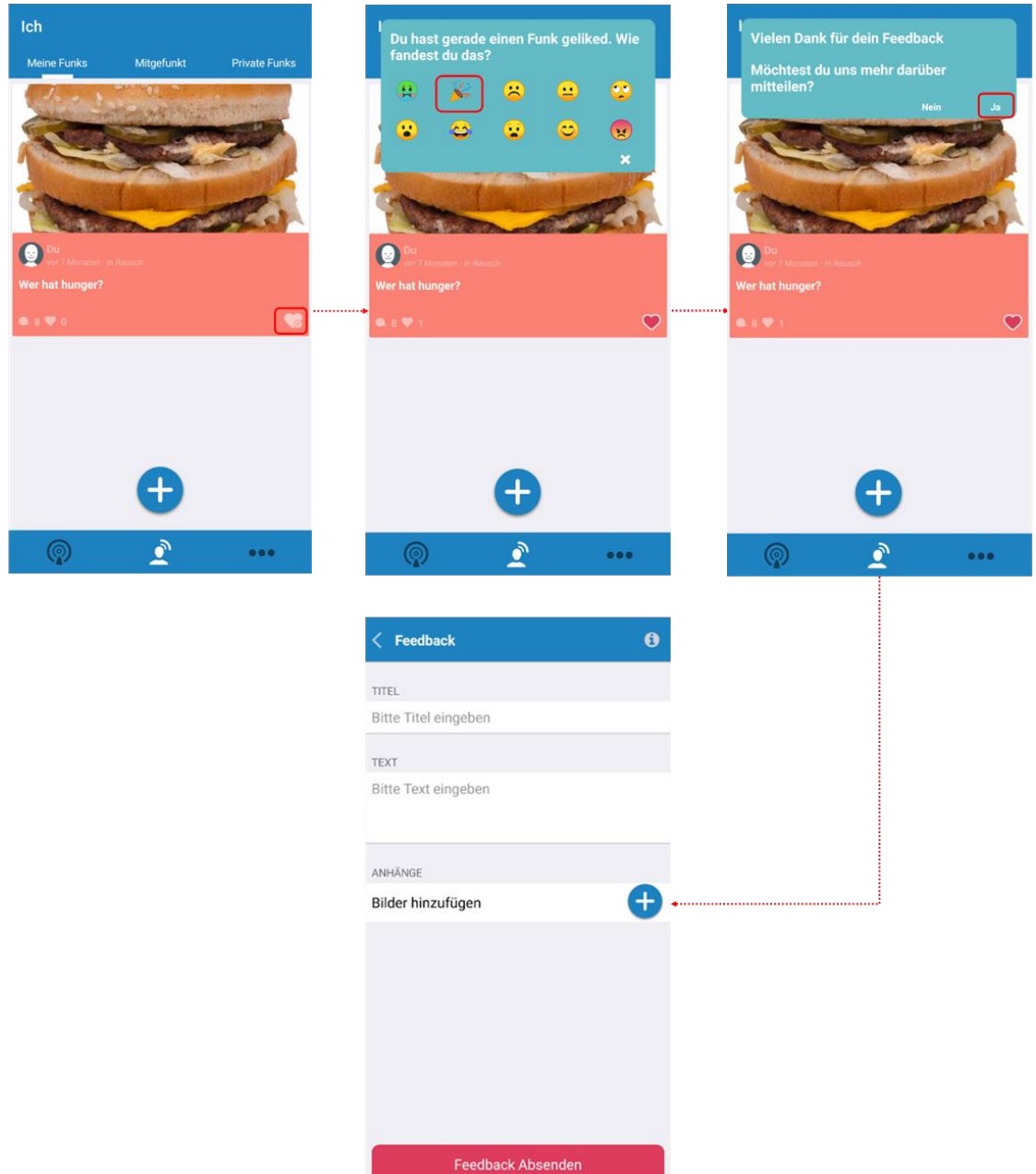

**Figure 11.** PfaffFunk pull feedback process. Top: Emoji can be selected. Bottom: Written feedback can be given, and a picture can be added.

We will continue reflecting on our procedures and example solutions critically and will adapt them in the future to come up with solutions for the future city district that will fit the people who will live and work there in the best possible way.

## 5. Perspectives on the Future

Since the construction work has not yet been completed and no user can yet be located in the city district, it is not possible for us to test the planned ecosystem in its real environment with real users yet. Since they have not been deployed yet, many applications are still in a development or prototype state. However, a number of adjustments and

further developments are still necessary before they can be used in a productive system. This includes the completion of the ICT platform. It already consists of a large number of applications, for example a chat functionality, a calendar, or gamification aspects. However, these need to be expanded with additional applications before the system goes live. The new ideas, which have currently been implemented as prototypes with the help of the mock platform, must be evaluated further. If they prove to be useful, they will be implemented and can be tested in the real environment. In addition, the MiniLautern game will be part of an exhibition so that it can be used by many people visiting the city district.

On our way towards creating a sustainable digital ecosystem for a smart city district in Kaiserslautern, we aim to accompany the entire innovation process for this future smart city district. Therefore, the purpose of or the conditions for some of our current solutions might change in the future or might even not be relevant anymore at a certain point in time as they have fulfilled their purpose.

Solutions that we think will work exactly in the same manner in 10 years from now are, for example, the mock platform as well as the Fish n' Tips app. Of course, there will be a process of iterative updates on them, such as Fish n' Tips obtaining different types of tips and the mock platform being updated for future services and frameworks, but their purpose will stay the same.

In contrast to this, we assume that the currently very important ICT concept will become less relevant, if not even irrelevant altogether, in the future. The reason for this is that the concepts being explained will become more and more common in future district planning. Similar to the concept of mobile computing, it will become widespread and therefore known to the public. Similarly, the need for having MiniLautern to bring innovative mobility concepts to the public mind will become obsolete as the upcoming district will feature this concept as part of its DNA. Therefore, people can see these concepts in action at least in this district.

In between are solutions that will serve a different purpose or will be organized differently in the future. For example, our district workshop will not be led by researchers anymore but rather by a community of digital enthusiasts driving the educational mission of the workshop. Moreover, PfaffFunk will move from informing the public about the project and its progress to a collaboration and communication solution for the people living in the district. Furthermore, the role of our map will change from explaining what digitalization means for a district to a more informative solution explaining what has been done in the district and how.

This knowledge leading to the insight that solutions might have to be altered, removed, or added to the ecosystem results in a continuous dialog with the stakeholders in all project phases to ensure that the current needs are addressed properly and future needs have already been identified.

## 6. Summary and Conclusions

In this article, we provided a framework for creating digital solutions for a smart city district. We shared procedures, example solutions, and lessons learned about what the digitalization of such a smart city district might look like. We provided insights from our research project about the vision and concepts we developed, our smart city district digital ecosystem, and dissemination and events. These three framework areas are those that we experienced as being highly relevant when thinking about the digitalization of a city district. Of course, we do not claim that these experiences support all other smart city projects, as such projects are highly individual, and each project has its own challenges. However, from our point of view, it provides a baseline to at least reflect on these areas also in other smart city projects and to decide what can be used and what does not fit in another context.

With respect to our four lead questions (LQ), we gained several insights throughout our activities, which we shared in Section 3 and discussed in Section 4.

- How to elicit needs and user requirements for digital solutions without knowing the users of the future city district? In order to understand the requirements of citizens living and working in the smart city district in the future, although no access to them is possible yet, we performed certain events such as hackathons. The platform we developed is a concrete instrument that was used to create prototypical solutions that might be used for future services. Furthermore, we used personas to anticipate requirements from fictional but realistic people (LQ1).

- How to communicate the vision, concepts, and future solutions without having actually implemented them—in other words, how to build a bridge from today to tomorrow and prepare users for the solutions? When communicating such solutions, we do, for example, prepare a digital ecosystem map with several different views on it. Our understanding is to make it as simple as possible and to arouse curiosity. Our initial concept was also created in a way that it is attractive to read the document. Furthermore, we came into contact with people at specific events such as the "Night of Science" (LQ2).

- How to deal with the challenge that needs will continue to change until the concrete implementation of digital solutions and usage by citizens in the future? That more change will happen that has an influence on our solutions is something we expect. This is also true for the people and their needs themselves. In our MiniLautern game, for example, we provide people with a situation in which they currently find themselves. In a solution like this, we can easily introduce new concepts and adapt the game so that future concepts—that we currently do not even think about—can play a role later. The workshop will be a place where new solutions can be thought of and developed, even if needs change over time (LQ3).

- How to test and evaluate the vision, concepts, ideas, and prototypical implementations without users of the future city district? Finally, we already tested some solutions with a limited number of participants, but we want to extend this heavily in the upcoming months. One example is the hackathon, which was used to test our mock platform and the solutions that were created and judged by a jury or other teams. The mock platform, but also the real ecosystem platform behind it, is created in an open manner so that other parties can later on use and test digital services (LQ4).

With respect to future work, we have shared many ideas in Section 5 already. In addition, we are currently planning a follow-up project that might start in October 2022. In such a successor project, we want to evaluate further solutions (not only our digital solutions, but also solutions from other partners in the project) and focus much more on feedback from citizens and other stakeholders such as planners, as well as gathering and analyzing data from different devices such as photovoltaics or smart light bulbs.

**Author Contributions:** Conceptualization, F.E., P.M., S.P., and P.S.; methodology, F.E., P.M., S.P., and P.S.; software, P.M., S.S., and P.S.; validation, F.E., P.M., and S.P.; investigation, F.E., P.M., S.P., and P.S.; writing—original draft preparation, F.E., P.M., S.P., S.S., and P.S.; writing—review and editing, F.E., P.M., S.P., S.S., and P.S.; visualization, F.E., P.M., and S.P.; supervision, F.E.; project administration, F.E. and P.M. All authors have read and agreed to the published version of the manuscript.

**Funding:** The research described in this paper was performed in the EnStadt:Pfaff project (grant no. 03SBE112D and 03SBE112G) of the German Federal Ministry for Economic Affairs and Energy (BMWi) and the Federal Ministry of Education and Research (BMBF).

**Institutional Review Board Statement:** Not applicable.

**Informed Consent Statement:** Not applicable.

**Acknowledgments:** We thank Sonnhild Namingha for proofreading.

**Conflicts of Interest:** The authors declare no conflict of interest. The funders had no role in the design of the study; in the collection, analyses, or interpretation of data; in the writing of the manuscript; or in the decision to publish the results.

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
