# Peer review of "Towards a Digital Ecosystem for a Smart City District: Procedure, Results, and Lessons Learned"

_smartcities, doi:10.3390/smartcities4020035_

Round 1

Reviewer 1 Report

Thank you for very interesting and promising paper. 

If this aims to be a research paper, I would suggest three following major changes:

  1. Increase the theoretical part of the paper. At this stage, there is no conceptual framework developed and there are no research-based questions raised.
  2.  Add a methods section describing research tools and  data collection.
  3. Make the project-related descriptive part much briefer.

If this does not aim to be a research paper then I am not a right person to evaluate it. 

Author Response

Dear reviewer,

First of all, thank you very much for reviewing our submitted publication and giving valuable feedback. We addressed your recommendations and explain in the following how we addressed them.

“Thank you for very interesting and promising paper.”

Thank you very much for the positive feedback! It was our aim to provide results from our research project that are interesting for the community, and we are also convinced that our shared results are relevant for other researchers, practitioners, and other readers.

“If this aims to be a research paper, I would suggest three following major changes: Increase the theoretical part of the paper. At this stage, there is no conceptual framework developed and there are no research-based questions raised.”

Thank you again for the point raised. We checked our paper again and changed several parts in the publication according to your suggestion. First, we introduced the term framework already in the abstract and in the introduction now, as this is a very good idea to hold together our whole work. Indeed, we also share the idea of a framework that we followed in our work; however, we also clarified that we did not follow a one-method-fits-all approach, but used several concrete procedures due to the applied research character of the project and the diverse topics we currently address. In Section 3, we also clarified that we explicitly divided our framework into the three parts vision and concepts, digital ecosystem, and dissemination and events. Within each category, we described our detailed procedures, which are part of the overall framework. Concretely, at the beginning of Section 3, where we introduced Figure 1 with our results, we added an explanation of how we understand the framework. Also, in Section 4, which is our discussion, we again explained how we understand the framework and how we applied it and what the concrete outcomes were.

Throughout the entire article, we followed the four lead questions we introduced in the introduction. They are our research questions, but the term “lead questions” fits the explorative and applied research characteristic of our environment better. However, in principle, they are research questions, and we answer them in detail in each subsection of Section 3, reflect on them again in each of the three summary and conclusion sections in Section 3, and summarize them again in Section 5 where we mention our main contributions.

“Add a methods section describing research tools and data collection.”

Thank you for this recommendation. In Section 4, we discuss what kinds of data we have been able to collect so far, and make it clear why it has been difficult for us until now to collect data from real citizens in our city district, as there are currently many construction work activities going on. In Section 3, we also emphasized where we were able to collect data so far, for example, at the booth during the Night of Science event, or in the evaluation of the MiniLautern game with participants. We hope that this clarifies this point for you.

“Make the project-related descriptive part much briefer.”

We went through the whole paper and checked this point. In Section 2.1, we reduced the description to a certain extent and removed the bullet point list. In Section 3, we identified several parts that could be shortened. As these are too many to list here, please have a look at the revised version of the paper.

We hope that we have addressed your recommendations adequately and would be happy to get further feedback from you if you see more improvement potential. Thank you again.

Reviewer 2 Report

The problem, which is considered in paper „Towards a Digital Ecosystem for a Smart City District: 2 Procedure, Results, and Lessons Learned“, is important in the research domain.

However, the paper need to be significantly improved in several domains:

  1. In the introduction section, connection between Digital Ecosystem and Smart City District should be more analyzed.

  1. The whole manuscript needs rework so writing style is in third person (singular or plural).

  1. The authors should provide translation of the figures that are presented in German.

  1. The authors have formulated the lead questions in a good manner.

  1. The section 2.1 is too long so it looks more like the dissemination of the project than literature review. The explanation of the project should be described by less words.

  1. The whole section 3 should get shorter.

  1. The section Conclusion needs to include main contribution of the research.

Author Response

Dear reviewer,

First of all, thank you very much for reviewing our submitted publication and giving valuable feedback. We addressed your recommendations and explain in the following how we addressed them.

“The problem, which is considered in paper „Towards a Digital Ecosystem for a Smart City District: 2 Procedure, Results, and Lessons Learned“, is important in the research domain.”

Thank you very much for the positive feedback! It was our aim to provide results from our research project that are interesting for the community and we are also convinced that our shared results are relevant for other researchers, practitioners, and other readers. We also identified the problem as being relevant and as one to which we can contribute, so that other researchers, in particular, can consider our contributions in their own smart city environments.

“However, the paper need to be significantly improved in several domains:

In the introduction section, connection between Digital Ecosystem and Smart City District should be more analyzed.”

Thank you for the reasonable comment. We checked the introduction again and refined our argumentation at the end of the second paragraph to make the connection between digital ecosystems and a smart city clearer.

“The whole manuscript needs rework so writing style is in third person (singular or plural).”

We are a bit unsure what exactly your suggestion is here. We used the “we” form throughout the paper as we created all the results by ourselves and wanted to give first-hand information and describe it in this form. We also looked at other already accepted publications of this special issue, for example, “Artificial Intelligence and Robotics in Smart City Strategies and Planned Smart Development” or “IoT-Enabled Smart Sustainable Cities: Challenges and Approaches”, and checked which form the authors used, and found that they used the same style. However, maybe you can clarify further what exactly you see as the improvement potential regarding the writing style so that we can incorporate this. Thank you.

“The authors should provide translation of the figures that are presented in German.”

Thank you for the hint. We changed the language, e.g., in Figure 7 (MiniLautern game). In the figures showing our concepts or the apps (Section 3.1.1, 3.1.2), no English version exists. However, to make up for this, we described in great detail what can be seen in the captions. In this case, the actual text in the figures is not relevant from our point of view, but the visualization and the concepts are, and we want to give these impressions to the reader.

“The authors have formulated the lead questions in a good manner.”

Thank you very much for the positive comment.

“The section 2.1 is too long so it looks more like the dissemination of the project than literature review. The explanation of the project should be described by less words.”

We reduced the description in this section and removed the bullet point list, summarizing the points very briefly instead.

“The whole section 3 should get shorter.”

In Section 3, we checked where we could shorten the descriptions and did so in several parts. For more details, please see the revised manuscript as the number of reductions in Section 3 are too many to list them here.

“The section Conclusion needs to include main contribution of the research.”

Thank you for the recommendation. We checked Section 3 again and ensured that at the end of each subsection, a conclusion is presented with respect to the addressed lead questions. We also renamed the final Section so that it is clearer that it is not only a summary, but also the conclusion with respect to the four lead questions, which comprise the main contributions of our work.

We hope that we have addressed your recommendations adequately and would be happy to get further feedback from you if you see more improvement potential. Thank you again.

Round 2

Reviewer 1 Report

Thank you again for this interesting paper. 

I agree this paper to be proceeded further. Nevertheless, it is not a rigid research article but a paper describing one smart city project making it more valuable as a contribution for practitioners 

Reviewer 2 Report

The authors have improved the manuscript.